

# A Halo abstraction for distributed n-dimensional structured grids within the C++ PGAS library DASH

Denis Hünich and Andreas Knüpfer

ZIH, Technische Universität Dresden, Dresden, Deutschland

## ABSTRACT

The Partitioned Global Address Space (PGAS) library DASH provides C++ container classes for distributed N-dimensional structured grids. This article presents enhancements on top of the DASH library to support stencil operations and halo areas to conveniently and efficiently parallelize structured grids. The improvements include definitions of multiple stencil operators, automatic derivation of halo sizes, efficient halo data exchanges, as well as communication hiding optimizations. The main contributions of this article are two-fold. First, the halo abstraction concept and the halo wrapper software components are explained. Second, the code complexity and the runtime of an example code implemented in DASH and pure Message Passing Interface (MPI) are compared.

## INTRODUCTION

### New trends in parallel and HPC programming

High performance computing (HPC) is an essential tool for challenging scientific and engineering simulations. It has been dominated by Message Passing Interface (MPI) (*MPI Forum, 2015*) and MPI-style parallelism for a long time. Today, the notion of MPI+X is also generally accepted practice to reach the highest scalability while efficiently using distributed-memory clusters comprised of multi-core or many-core cluster nodes.

The partitioned global address space (PGAS) concept is an alternative to the message passing concept. It provides random memory access between many processes in a parallel application running across many distributed-memory nodes. Remote access is still much slower than local access, but can reduce the complexity of distributed-memory parallel programming with little performance penalties. MPI also adapted PGAS in the form of one-sided communication operations with the MPI 3 standard.

### Spacial domain decomposition

This work aims at the large group of parallel HPC codes doing some form of computation on a spacial simulation domain with a domain decomposition to scale up for high parallelism with either strong or weak scaling. In contrast, a functional decomposition cannot scale because there is only a fixed number of different operations to perform.

Corresponding author
Denis Hünich, denis.huenich@tu-dresden.de

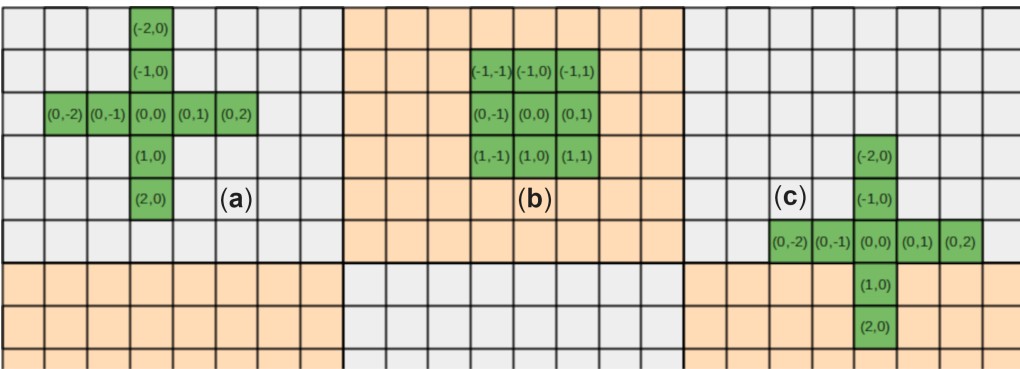

**Figure 1** Two shapes of a nine-point stencil: (A) left: center ±2 stencil points in both horizontal and vertical direction, (B) middle: center ±1 stencil point in each direction, and (C) right: the first stencil crossing the partition boundaries.

In the course of discretization the simulation domain can be mapped to a structured or unstructured grid. A structured $n$-dimensional grid has a regular neighbor relationship between grid elements that can be directly mapped to the storage order of $n$-dimensional arrays. Because the neighbor relation does not need to be stored explicitly, grid elements can be accessed more efficient and data structures are simpler in its design. An unstructured grid allows arbitrary neighbor relations to represent more flexible simulation domain geometries. However, it requires to store the neighbor relationship explicitly and so an indirect access of grid elements, *i.e.,* accessing the neighbor reference before the actual neighbor element. In this work we focus on structured grids only, albeit a generalization of the following is also of interest for unstructured grids and shall become the focus of future work.

## Structured grids, stencil operations, partitions, and halo areas

Simulations with regular structured grids often use stencils to describe surrounding grid elements (neighbors) relative to the current grid element (center) for basic calculation steps to update/modify the center element. Neighbors and often the center itself are called stencil points and their arrangement is named stencil shape. Figure 1 shows two examples of a two-dimensional nine-point stencil. Although both stencils have the same number of neighbors their shape is different. Fig. 1A has up to ±2 neighbor elements in every dimension, while Fig. 1B uses all direct neighbors.

In case a grid is to big to fit into the memory of one compute node, the grid needs to be divided into partitions which are distributed across many compute nodes (distributed memory). Elements on the boundary of a partition with neighbors on other partitions (boundary elements) can access these neighbors only remotely *via* communication substrates, such as MPI. Doing this element-wise is most inefficient, because the data of each neighbor has to be requested and transferred over the network with a much higher latency than a local memory access. Unless it is used occasionally (few remote accesses), this will drastically slow down the entire parallel application.

To avoid this performance decrease, "halo areas" (*Kjolstad & Snir, 2010*) that contain local copies of all required neighbor elements located on remote partitions can be used. The number and size of halo areas depend on the shape of the applied stencils, the number of neighbor partitions and the size of the partitions itself (number of dimensions and distribution pattern). The halo elements are copied from their remote original values with few bulk transfers instead of many individual transfer requests and then be accessed locally. Halo updates need to be done in a phase of the parallel algorithm where the necessary remote values are up-to-date and ready to be transferred.

This procedure significantly increases the performance. Furthermore, phases of purely local computations (inner elements), no stencil point needs remote resp. halo elements, can be overlapped with asynchronous halo transfers. Afterwards, all remaining elements (boundary elements) can be updated. If the halo transfers are finished while the inner elements are still computed, no additional waiting time is required. Most likely, this is the case when the inner area of a partition is much larger than the boundary area.

# RELATED WORK

## Related work regarding C++ parallelization abstractions

We identified three different approaches adopting the PGAS concept for parallel HPC programming:

| | |
|---|---|
| PGAS language extensions and separate languages | UPC++ (*Zheng et al., 2014*) and Co-Array C++ (*Eleftheriou, Chatterjee & Moreira, 2002*) are designed as C++ language extensions, whereas Chapel (*Chamberlain, Callahan & Zima, 2007*) and X10 (*Charles et al., 2005*) are separate PGAS programming languages. |
| Libraries with parallel programming APIs | The most dominant one, MPI (*MPI Forum, 2015*), also adopted PGAS operations (calling it "one-sided communication"). GASPI (*Grünewald & Simmendinger, 2013*) and OpenSH-MEM (*Chapman et al., 2010*) are alternative libraries realizing the PGAS concept. |
| C++ libraries | DASH (*Fürlinger et al., 2014*) (Section 'The C++ template library DASH and its NArray container'), HPX (*Kaiser et al., 2014*), Kokkos (*Edwards, Trott & Sunderland, 2014*) and STAPL (*Buss et al., 2010*) are libraries that provide communication APIs together with abstractions for distributed data structures. The HPX project addresses distributed memory systems, but doesn't support n-dimensional containers. Kokkos also provides multi-dimensional containers, but focuses on shared memory systems only. STAPL shares concepts like local views on data and representation of distributed containers with DASH, but seems to be a closed source project and doesn't aim at classical HPC applications. None of the mentioned PGAS approaches offer stencil and halo abstractions for n-dimensional data containers. |

### Related work regarding halo exchange mechanisms

The basic concepts of halo areas (also called "ghost cells") and boundary data exchanges are presented in _Kjolstad & Snir (2010)_ and are used in this work and other related approaches. The STELLA project (_Gysi et al., 2014_) provides a domain-specific embedded language using generic programming in C++ and supports stencil codes on structured grids by using OpenMP and CUDA. Compared to the presented approach it is limited to shared memory systems only. DUNE (_Bastian et al., 2008_) and PETSc (_Balay et al., 1997_) are both modular C++ libraries for partial differential equations using grid-based methods and sparse matrix computations. Using MPI, both projects can be used for distributed memory systems. ScaFES (_Flehmig, Feldhoff & Markwardt, 2014_) also uses MPI to distribute structured grids and to update halo areas. DUNE, PETSc, and ScaFES integrate the halo functionality as part of their solver frameworks instead of a generic and separately usable concept as presented here. ScaFES is closest to our approach and was designed to solve simple numerical methods like the explicit finite difference, whereas PETCs and DUNE are very complex frameworks and designed for more general purposes. Like ScaFES, our approach is designed for simple numerical methods that can be implemented fast and efficient. None of the mentioned libraries uses the concept of explicit local and global data accesses such as the DASH data containers.

## THE C++ TEMPLATE LIBRARY DASH AND ITS NARRAY CONTAINER

DASH is a data structure oriented C++ template library (_Fürlinger et al., 2014_) offering PGAS-like data container classes such as n-dimensional arrays, lists, or unordered maps for distributed-memory parallel applications. The elements in these containers can be accessed by local and global iterators. The iterator concept and other concepts in DASH follow the rules of the C++ Standard Library (_SL_) and are compatible with its algorithms.

To better understand how these containers work, we describe the structure of the n-dimensional array container (NArray) in the following. An NArray distributes $n \leq N$ parallel processes over an n-dimensional structured grid. Each process allocates a local memory block with a sufficient size. For example, a three-dimensional NArray with an extend of 10 in every dimension contains 1,000 elements. With $n = 4$ processes, each process has to allocate and manage memory for 250 elements of the given element type. The elements can be iterated in a logical global order (row major by default) with global iterators, shown in Listing 1 (top). Each process steps through all elements of the NArray; local and remote ones. All remote accesses require _inefficient_ data transfers, realized by the underlying DART transport abstraction library (_Zhou et al., 2014_). This is a lightweight PGAS runtime library, managing one-sided put/get communication in a blocking or asynchronous mode as well as local and global synchronization mechanisms. Currently, DART can be used with MPI (_MPI Forum, 2015_) or GASPI (_Grünewald & Simmendinger, 2013_) as its communication substrate.

Internally, a distribution pattern, provided by DASH or user defined, maps between logical indices (global and local) and the memory. Depending on the specified pattern the

memory of each process is mapped to one or more logical n-dimensional blocks of the grid. If only one block per process is used, all local elements are stored in a contiguous order following the given global order. Otherwise, the local elements might not be stored contiguously in the memory, despite their logical contiguous order.

Every process can iterate its *local* elements in a very fast and efficient way; shown in Listing 1 (bottom). DASH's local and global iterators can be used with standard for-loops, range based loops and algorithms of the *SL*. However, global iterators should be used carefully or for debugging purposes only. Local iterators should be preferred wherever possible.

```
1   // global iterator access
2   auto it_end = my_narray.end();
3   for(auto it = my_narray.begin(); it != it_end; ++it) {
4     std::cout << *it << "␣";
5   }
6   // local iterator access
7   for(const auto& elem: my_narray.local) {
8     std::cout << elem << "␣";
9   }
```

Listing 1: Element access via global and local DASH iterators.

### Using DASH NArrays for stencil operations

The DASH NArray is well suited for distributed structured compute grids and it is possible to use stencil-like accesses on the elements. However, the following major two issues prevent a simple implementation. First, the advice to use local iterators wherever possible works well for all inner elements (stencil needs no remote element access—shown in Fig. 1A). The number of inner elements can differ through different sizes of the local partitions and stencil shapes. This requires additional and possibly error-prone memory position calculations and is almost as complicated as corresponding MPI codes. The second and more severe issue is the management of boundary elements (at least one stencil point needs remote access—shown in Fig. 1C). Even for relatively few border elements, each individual remote access results in a performance loss. Managing a halo environment and halo updates is again not a simplification for users. In the next section, we present the concept and implementation of a halo environment which solves both issues in a convenient and highly efficient manner. It covers varying inner and border areas in multiple dimensions, efficient asynchronously halo updates (in the background while doing other computations), and support for multiple stencils applied to the same NArray. Furthermore, grid extensions or the number of dimensions can be changed with small effort.

## HALO ENVIRONMENT FOR THE DASH NARRAY

As a solution to the aforementioned challenges we designed and implemented the *HaloWrapper* abstraction for DASH on top of the NArray container class (but deliberately not as a replacement) and evaluated it. We focused on (1) access of stencil points with a tailored iterator (center or neighbor elements), (2) consistent local access to all stencil points (either inner or halo areas), and (3) built-in asynchronous halo updates allowing perfect

communication overlap. In the following we describe essential parts of the *HaloWrapper* design.

### *HaloWrapper* specification

Besides the NArray, other essential components for the *HaloWrapper* are stencil definitions and preferences for the global grid border.

A stencil is defined with the class StencilSpec, a collection class managing all stencil points belonging to this stencil. Each stencil point, represented by the class StencilPoint, has a weight (coefficient) and coordinates relative to the center. Listing 2 shows the definition of the stencil described in Fig. 1B. The expression StencilPoint $< 2 >$ defines stencil points for two dimensions only. StencilSpec $<$StencilT, 8 $>$ expects eight stencil points of the type StencilPoint $< 2 >$. Each stencil point is constructed with two arguments, which point to the direction and distance of the neighbor in each dimension. For example, StencilT($-1,-1$) creates a StencilPoint pointing to the neighbor one element before the center in each dimension (north west) and the default coefficient of 1.0 In case a different coefficient shall be used, a third argument needs to be passed.

```
1  using StencilT       = dash::halo::StencilPoint<2>;
2  using GlobBoundSpecT = dash::halo::GlobalBoundarySpec<2>;
3  dash::halo::StencilSpec<StencilT,8> stencil_spec(
4    StencilT(-1,-1), StencilT(-1, 0), StencilT(-1, 1), StencilT( 0,-1),
5    StencilT( 0, 1), StencilT( 1,-1), StencilT( 1, 0), StencilT( 1, 1))
     ;
6  GlobBoundSpecT bound_spec(BoundaryProp::CYCLIC, BoundaryProp::CYCLIC);
```

Listing 2: Stencil specification for a two dimensional full stencil of width 1. The center element doesn't need to be specified explicitly.

Additionally, the *HaloWrapper* provides global boundary condition specifications for a distributed compute grid in the sense of PDE boundary conditions. It is specified *via* the class GlobalBoundarySpec (see Listing 2 line 6) and provides three different conditions for each dimension separately. The default setting NONE creates no halo areas at outside edges of the global grid. Because data requests of these neighbors cannot return valid data, stencil operations are excluded for these border areas. The alternative setting CYCLIC logically connects the global boundaries by pointing to the opposite side of the global grid in this dimension. The setting CUSTOM offers a convenient way to provide arbitrary values for those halo elements, either in a static (only set once) or dynamic way (updated between the iterations in any way the program sees fit). Such halo regions get excluded from the built-in halo updates.

### Internal management of halo partitions

The local memory portion of each NArray process, in the following named partition (see the 'Structured grids, stencil operations, partitions, and halo areas' section), is wrapped by the HaloWrapper. To identify all neighbor partitions around the local one, we developed the region concept. Each partition can have $3^d$ regions for a $d$-dimensional NArray, which are identified by a unique index and unique coordinates. Each region has also an extent that defines its size. Figure 2A illustrates all regions for two dimensions; the center region has the index 4 and the coordinates (1,1). The other regions represent partitions managed by other

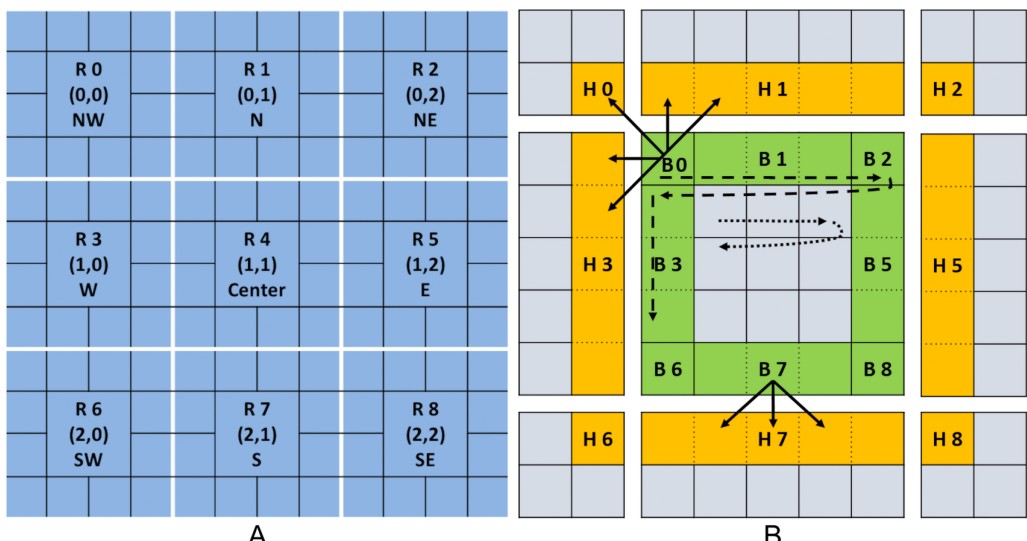

**Figure 2** Region concept for a two-dimensional partition: (A) A center partition and all surrounding neighbor partitions. (B) The regions inside a single partition with HaloRegions (yellow), BoundaryRegions (green) and the InnerRegion in the middle.

processes. The region index follows the row major linearization (last region coordinate grows fastest) and can be easily mapped to the coordinates and *vice versa*. Depending on different factors such as the partition's location in the grid, the number of processes or the BoundaryConditions, the number of actually used regions may be smaller than $3^d$.

Besides the mapping of regions to partitions, the HaloWrapper creates another two types of regions; named HaloRegion and BoundaryRegion, shown in Fig. 2B. The region concept stays the same but the mapping differs to one described before (see Fig. 2A). HaloRegions are located on the same partitions as in Fig. 2A, but only mark a subset of elements, that are necessary for halo updates. BoundaryRegions are located on the local partition only and mark elements that need at least one halo element for a stencil operation. The benefit of HaloRegions is quite obvious, but why create extra BoundaryRegions? Figure 2B shows two stencil operations with a different number of halo accesses (one centered in B0, the other centered in B7). A stencil operation on the edges (*e.g.*, B7) needs finished halo updates for one HaloRegion only, while stencil operations near a corner (*e.g.*, B0) need at least for three finished halo updates (three HaloRegions). Therefore, BoundaryRegions allow to process elements on the boundary depending on the halo update status.

Figure 2B shows all HaloRegions (yellow) and BoundaryRegions (green) for a local partition with a full ±1 stencil on a two-dimensional grid. While the center HaloRegion is irrelevant, the center BoundaryRegions represents all inner partition elements that do not need any halo element access for the stencil operation. For convenience we call this region InnerRegion in the following. The sizes of the HaloRegions and BoundaryRegions result from all stencils passed to the HaloWrapper (region extent) and the partition extension for the remaining dimensions. *e.g.*, H3 in Fig. 2B has an extent of five in the first dimension and an extent of 1 in the second dimension (stencil width of 1). If multiple StencilPoints

have the same direction (*e.g.*, multiple StencilSpecs), the region extent is determined by the maximal width of all these StencilPoints.

## Internal management of halo memory and halo data exchanges

The copies of all HaloRegions elements, in the following called halo elements (Section 'Structured grids, stencil operations, partitions, and halo areas', are stored in one separate contiguous memory block named HaloMemory. Elements of the first HaloRegion (ID 0) are stored at the beginning, followed by the next HaloRegion (ID 1) until the last HaloRegion (ID $3^{dimensions} - 1$). A virtual layer maps the HaloMemory to the assigned HaloRegions to provide region-wise halo element access. Because HaloRegion elements are just a segment of a remote partition, they are rarely available contiguously and must be transferred with multiple communication requests. Figure 3 shows two corner HaloRegions for a two- and three-dimensional partition. While the two-dimensional HaloRegion (Fig. 3A) needs two communication requests, the three-dimensional (Fig. 3B) already needs four. The number of communication requests for corner elements is negligible, compared to a one element thin layer of a three dimensional partition, where each halo element needs a communication request. DART supports strided communication requests for these kind of scenarios, but it is up to the communication substrate for how to handle these requests. While MPI works fine with big strided communication requests, GASPI does not. Therefore, to support both communication substrates and still efficiently update the requested halo elements, we decided to buffer the HaloRegion elements in a contiguous memory block first and afterwards transfer the buffered data with one communication request. This requires more memory but results in lower waiting latencies (finalizing all created communication requests).

The *HaloWrapper* supports two kinds of halo element updates; blocking and asynchronous. The blocking halo update call returns when all halo updates are finished and does not need additional synchronization methods (*e.g.*, barriers). The asynchronous counterpart initiates the halo updates and returns immediately (Listing 3 line 9). An additional wait function ensures that all or only a subset of the halo updates are locally finished and the halo elements are ready to be used (Listing 3 line 17).

To guarantee that the HaloRegion elements are up-to-date, all involved partitions have to be synchronized. This can be achieved explicitly with a global barrier at the end of each iteration, but forces each partition to wait for the slowest one. To avoid this possible performance bottleneck we implemented a signaling environment that synchronizes a subset of partitions only. The signal handling is part of the halo update process; initiation and waiting. The initiation (Listing 3 line 9) is divided into five parts (1-5), while waiting (Listing 3 line 17) consists of two parts (6-7):

(1) Check if all neighbor partitions finished the data transfer of all buffered data. If not, wait until it is done.
(2) Buffer all local partition elements required by neighbor partitions.
(3) Send a signal to all neighbor partitions that updated data are available now.
(4) Check and maybe wait until all HaloRegions are ready to be requested
(5) Initiate the asynchronous halo update.

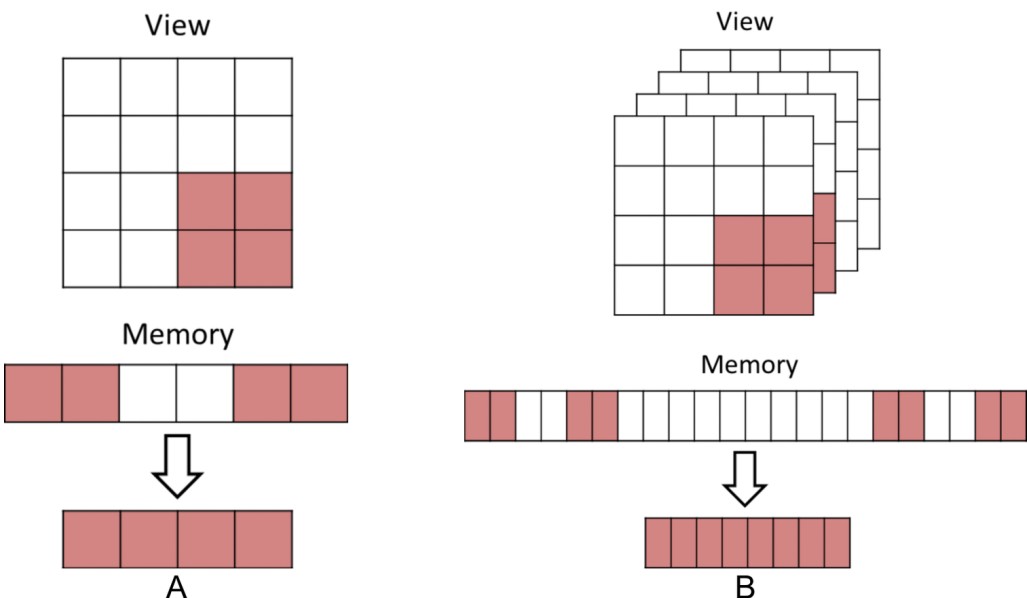

**Figure 3** Halo data exchange: Remote strided data to contiguous memory: (A) Corner HaloRegion only has one fixed offset between data chunks (B) Corner HaloRegion already has two different offset between data chunks for three dimensions.

(6) Wait until all or a subset of the requested HaloRegions updates are finished.

(7) Send signals to all neighbor partitions, that the halo update process is done.

The signals itself are one-sided communication calls that modify a DASH Array allocated for this purpose only and is build with the same team as the wrapped NArray. Each partition manages the aforementioned signals for each direct neighbor itself. The signals used in (1) and (7) are necessary especially for smaller grids, in case a partition starts to modify buffered halo elements, while neighbor partitions still requesting the data from the buffer. If this is not the case, these two signal steps can be disabled.

### Stencil point access

To access grid elements itself, the provided NArray iterators are sufficient. But as stated in Section 'Using DASH NArrays for stencil operations' these iterators do not support a proper stencil access. Therefore, the HaloWrapper provides three different iterator types that support neighbor element access in form of an extra method *via* a specific stencil point ID (Listing 3 line 18 and 19) or a StencilPoint instead. The iterator types differ in the covered area of a partition and the way they access the requested neighbor data.

**StencilIterator**      iterates over all partition elements that have valid neighbors (depending on the GlobalBoundarySpec, see Section 'HaloWrapper specification'). A neighbor access automatically resolves the origin of the accessed element (HaloMemory or partition element) and afterwards returns the data.

| | |
|---|---|
| **InnerIterator** | iterates elements within the InnerRegion. Because none of the accessed neighbors requires HaloMemory access, location checks, as used in the StencilIterator, are not necessary. The iteration order is taken from the memory order specified by the wrapped NArray (row or column major). The InnerRegion represents a sub-partition only and can not be accessed contiguously. That means the InnerIterator needs to reevaluate the memory position each time it is moved. Because the pre-increment operator is often used, we optimized its positioning. The coordinates, necessary for the current position in the partition, are not recalculated each iteration. Instead, only necessary parts of the coordinates are increased. The neighbors always have a fixed offset relative to the center and so can be easily calculated (fixed offset). The dotted arrows in Fig. 2B show exemplary the iteration path of an InnerIterator. |
| **BoundaryIterator** | covers all identified BoundaryRegions. Their number depends on the StencilSpec and the GlobalBoundarySpec (Section 'HaloWrapper specification'). Inside a BoundaryRegion the iteration order is equivalent to the memory order of the wrapped NArray. If the BoundaryIterator passes the last element of a BoundaryRegion, the first element of the next BoundryRegion becomes the new center. BoundaryRegions are traversed from the smallest to the largest BoundaryRegion ID. An example of the iteration path over four *BoundaryRegions* is shown in Fig. 2B with dashed arrows. |

If the HaloWrapper is build with multiple StencilSpecs that differ in distance or direction, the BoundaryRegions including the InnerRegion are sized for the maximal stencil widths. To use these as basis for the Inner and BoundaryIterator for different stencils is not optimal, because some partition elements are accessed by the BoundaryIterator instead of the faster InnerIterator. For this reason the HaloWrapper provides so called StencilOperators (*e.g.*, in Listing 3 line 7) that adapts to one specific StencilSpec. Each StencilOperator provides StencilIterators adjusted to the given StencilSpec and additional methods to simplify stencil operation tasks.

To also support an alternative access, besides iterators, we also implemented a coordinate based access strategy (*e.g.*, element_access[-1][0]) that can access inner and boundary regions too. Here, the user needs to take care of the coordinate ranges itself to not access invalid memory. The use of iterators is recommended, because they do not require extra coordinate management and do work with the C++ Standard Library algorithms.

Listing 3 shows a snippet of a simple two-dimensional heat equation to demonstrate the concepts and features described before by following the basic halo concept described in Section 'Structured grids, stencil operations, partitions, and halo areas'. The first two lines specify a five-point stencil (no stencil point for the center necessary) in form of a StencilSpec. The global borders are connected (line 4). Therefore the property *CYCLIC*

needs to be set for both dimensions. The next step is the creation of an HaloWrapper object (line 5). To show that multiple StencilSpecs can be passed to the HaloWrapper, a fake StencilSpec named s_spec_2 is passed. A StencilOperator for the StencilSpec *s_spec* is requested and stored in line 7. The for-loop from line 9 to line 30 contains the update process of the grid for each turn and includes four steps to be executed collectively for all distributed partitions. First, the halo updates are started asynchronously (line 10). Second, a stencil operation is performed on all InnerRegion elements with an InnerIterator provided by the StencilOperator (line 12 to 21). Third, waiting for all halo updates to be finished (line 23). Most likely, all halo updates are finished during the second step (depending on the InnerRegion's size). Finally, all BoundaryRegion elements are processed (from line 25 to 29). The loop looks similar to the InnerRegion (second part), but uses the BoundaryIterator instead.

```
1   StencilSpecT s_spec( StencilT(-1, 0), StencilT(1, 0),
2                        StencilT( 0,-1), StencilT(0, 1));
3   // Periodic/cyclic global border conditions for both dimensions
4   GlobBoundSpecT bound_spec(BoundaryProp::CYCLIC,BoundaryProp::CYCLIC);
5   HaloWrapperT halo_wrapper(src_matrix, bound_spec, s_spec, s_spec_2, \
        ldots);
6   // Stencil specific operator for a specific stencil_spec
7   auto stencil_op = halo_wrapper.stencil_operator(s_spec);
8
9   for (auto $i=0$; i < iterations; ++i) {
10    halo_wrapper.update_async(); // start asynchronous halo update
11
12    // Calculation of all inner elements (InnerRegion)
13    auto iend = stencil_op.inner.end();
14    for(auto it = stencil_op.inner.begin(); it != iend; ++it) {
15      // it.value_at(0) accesses neighbor represented by StencilT(-1,
            0)
16      auto center = *it;
17      double dtheta =
18      (it.value_at(0) + it.value_at(1) - 2 * center) / (dx*dx) +
19      (it.value_at(2) + it.value_at(3) - 2 * center) / (dy*dy);
20      ...
21    }
22
23    halo_wrapper.wait(); // Wait until all halo elements are updated
24
25    // Calculation of all boundary region elements
26    auto bend = stencil_op.boundary.end();
27    for(auto it = stencil_op.boundary.begin(); it != bend; ++it) {
28      //same as the inner part
29    }
30  }
```

Listing 3: Example of a simple 2D heat equation iteration loop with halo exchange and stencil operations

## EVALUATION

To evaluate the HaloWrapper we implemented a simple heat equation with a Jacobi solver for two- and three-dimensional (2D and 3D) structured grids. The global grid borders are connected (CYCLIC). The Jacobi solver requires two equivalent compute grids, one as read-only source grid and the other as write-only destination grid. Both grids alternate between each iterations. The stencil operations use a five-point-stencil for 2D and a seven-point-stencil for 3D with an offset of $\pm 1$ in every dimension. All grid elements are of type double precision floating point.

**Table 1  Lines of code (LOC) comparison of MPI and DASH for a 2D and 3D heat equation simulation in toal and for selected code phases.** All LOC are counted without comments and blank code lines. The value in brackets are LOC that had to modified and also include added LOC.

| Implementation | 2D | | | 3D | | | Δ LOC 2D to 3D | |
|---|---|---|---|---|---|---|---|---|
| | MPI | DASH | Δ LOC | MPI | DASH | Δ LOC | MPI | DASH |
| Total | 249 | 134 | 115 | 318 | 140 | 178 | 69 (132) | 6 (20) |
| *async* and *wait* | 20 | 2 | 18 | 28 | 2 | 26 | 8 (18) | 0 (0) |
| *inner* | 8 | 7 | 1 | 11 | 8 | 3 | 3 (6) | 1 (2) |
| *bound* | 18 | 7 | 11 | 36 | 8 | 28 | 14 (30) | 1 (2) |

An iteration is divided into the asynchronous update of the halo elements (*async*), the calculation of all inner elements (*inner*), waiting for the halo transfers to finish (*wait*), and the calculation of all boundary elements (*bound*). The *inner* part should overlap the halo update process started in *async*, and so minimize *wait*. The number of iterations is fixed to 100 (*calc*).

We implemented the aforementioned scenario in plain MPI and in DASH with the HaloWrapper. Additionally, we compared the performance of DASH with ScaFES, to validate the results with another abstraction layer.

## Source code comparison

One aspect to consider when writing in abstraction is to reduce the code complexity. It is quite subjective to define what is complex and what is not, we concentrated on the used lines of code (LOC). Also, we compared the LOC for different scenarios (2D and 3D) to show the necessary adaption effort.

The comparison of LOC has also a subjective part and it could be argued that certain parts can be hidden in extra functions, to result in less LOC. Still, the logic of these functions needs to be implemented and adapted for other use cases. Even if it is not a strict criterion for complexity, fewer or significant fewer LOC for an implementation lead to less code complexity and opportunities for inconsistencies. Also, a strong indication for less complexity and better maintainability is a minimal increase of LOC when transforming the code to higher dimensions, *i.e.*, 2D to 3D.

Table 1 shows the LOC for the MPI and DASH based implementations in total and for selected code phases (used in the performance measurements). As shown in the column Δ LOC for 2D and 3D DASH requires much less LOC to solve the problem. The only exception is the *inner* part, here MPI is similar to DASH. The major difference between MPI and DASH is the effort to adapt the 2D implementation to 3D. DASH needs six new LOC and has to modify 20 LOC in total, which is significant less than MPI.

Listing 4 shows the 3D version of the 2D example shown in Listing 3. The complete code examples can be found on GitHub (https://github.com/dash-project/dash-apps/tree/master/heat_equation).

```
1  StencilSpecT s_spec( StencilT(-1, 0, 0), StencilT(1, 0, 0),
2                       StencilT( 0, -1, 0), StencilT(0, 1, 0)
3                       StencilT( 0, 0, -1), StencilT(0, 0, 1));
4  // Periodic/cyclic global border conditions for both dimensions
5  GlobBoundSpecT bound_spec(BoundaryProp::CYCLIC, BoundaryProp::CYCLIC,
```

```
6                          BoundaryProp::CYCLIC);
7    HaloWrapperT halo_wrapper(src_matrix, bound_spec, s_spec, s_spec_2, \
       ldots);
8    // Stencil specific operator for a specific stencil_spec
9    auto stencil_op = halo_wrapper.stencil_operator(s_spec);
10
11   for (auto $i=0$; i < iterations; ++i) {
12     halo_wrapper.update_async(); // start asynchronous halo update
13
14     // Calculation of all inner elements (InnerRegion)
15     auto iend = stencil_op.inner.end();
16     for(auto it = stencil_op.inner.begin(); it != iend; ++it) {
17       // it.value_at(0) accesses neighbor represented by StencilT(-1,
          0)
18       double dtheta =
19         (it.value_at(0) + it.value_at(1) - 2 * center) / (dx*dx) +
20         (it.value_at(2) + it.value_at(3) - 2 * center) / (dy*dy) +
21         (it.value_at(4) + it.value_at(5) - 2 * center) / (dz*dz);
22       ...
23     }
24
25     halo_wrapper.wait(); // Wait until all halo elements are updated
26
27     // Calculation of all boundary region elements
28     auto bend = stencil_op.boundary.end();
29     for(auto it = stencil_op.boundary.begin(); it != bend; ++it) {
30       //same as the inner part
31     }
32   }
```

Listing 4: Example of a simple 2D heat equation iteration loop with halo exchange and stencil operations

## Performance comparison

In HPC an abstraction to simplify the coding is most useful if the performance remains. Therefore, we compared both implementations (plain MPI and DASH) in a weak and a strong scaling scenario. The runtimes were measured on the Bull HPC-Cluster "Taurus" at ZIH, TU Dresden on the *Haswell* and the *Romeo* partition. *Haswell* provides compute nodes with two Haswell E5-2680 v3 CPUs at 2.50 GHz (12 physical cores each) and 64 GB memory. Compute nodes on *Romeo* have two AMD EPYC CPU 7702 (64 physical cores each) and 512 GB memory. Hyper-Threading (*Haswell*) and Simultaneous Multithreading (*Romeo*) were disabled. The implementations were compiled with gcc 10.2.0 and used OpenMPI 4.0.5.

### DASH vs. MPI

The essential runtime contributions are *async*, *inner*, *wait*, *bound* and the total calculation time (*calc*). Each combination was measured three times and each plot shows the mean of these measurements. While *calc* is measured once per run, the other parts were recorded per iteration and per process. The mean of it was summed up to better compare their runtime share with *calc* in the plots. The weak scaling scenario increases the number of grid elements proportional to the number of compute nodes. Nearly the total memory of each compute node were used. The figures marked with (a) show the runtimes of the phases *calc*, *inner* and *bound*, while figures marked with (b) show the runtimes of the phases *async* and *wait*. The values plotted in the figures are not stacked.

Figures 4 and 5 show the results of the 2D heat equation on *Haswell* and *Romeo*. Both implementations show nearly identical runtimes (Fig. 4A) on *Haswell*, while on *Romeo*

PeerJ Computer Science ___________________________________________

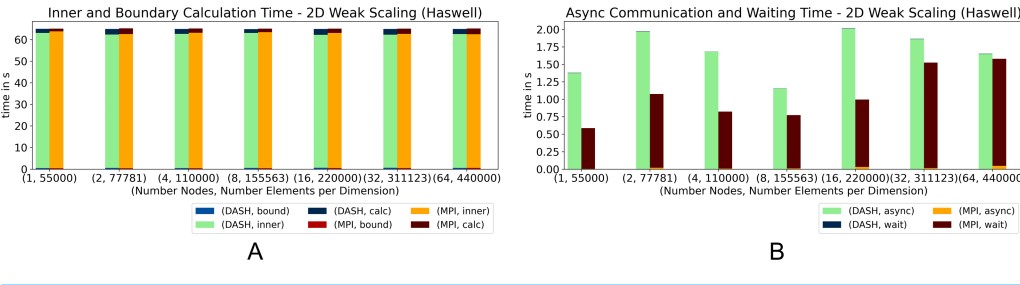

**Figure 4** (A–B) Weak scaling for 2D heat equation—DASH *vs.* MPI on Haswell (values not stacked).

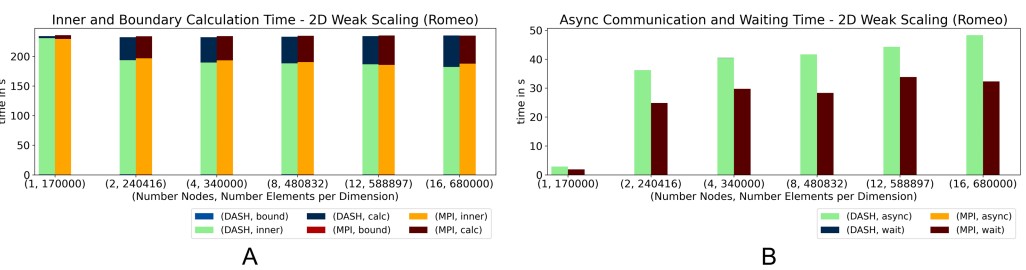

**Figure 5** (A–B) Weak scaling for 2D heat equation—DASH *vs.* MPI on Romeo (values not stacked).

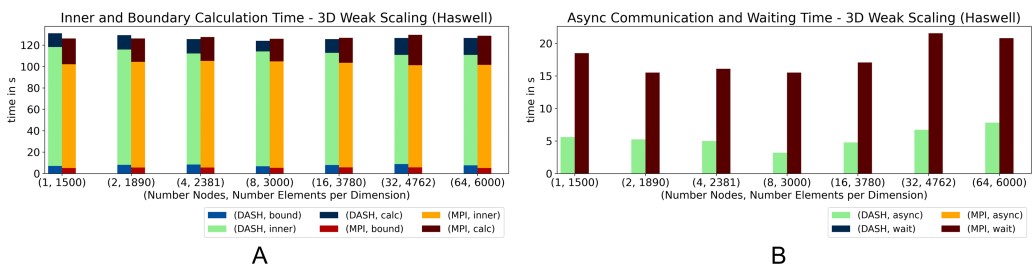

**Figure 6** Weak scaling for 3D heat equation—DASH *vs.* MPI on Haswell (values not stacked).

DASH performs slightly better (Fig. 5A). Interestingly, the runtime of the phases differ. While DASH calculated the inner elements faster, MPI handled the halo updates better. DASH needed compared to MPI almost no runtime in *wait*, therefore *async* behaved *vice versa*. The preparation of the halo data in DASH described in the 'Internal management of halo memory and halo data exchanges' section slowed down the *asyn* phase. In total DASH is slightly faster.

The results of the 3D pendant, Figs. 6 and 7, show that DASH and MPI performed very similar. On *Haswell*, DASH could not compete with MPI for one and two nodes, but outperformed MPI for four to 64 nodes. On *Romeo*, DASH and MPI performed quite similar. However, the contributions of some components changed entirely. MPI took more

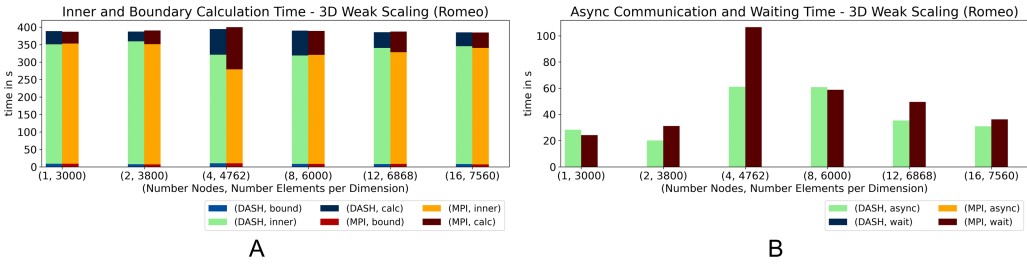

**Figure 7** **Weak scaling for 3D heat equation—DASH *vs.* MPI on Romeo (values not stacked).**

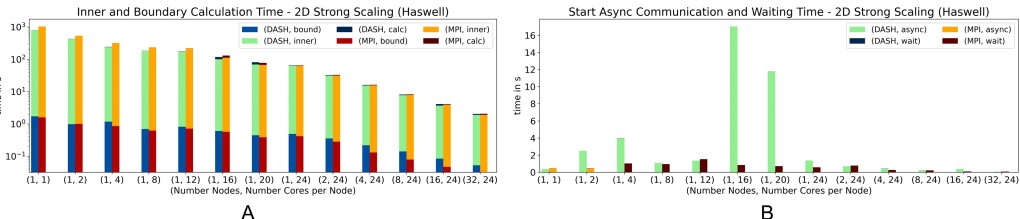

**Figure 8** **Strong scaling for 2D heat equation—DASH *vs.* MPI on Haswell (values not stacked).**

time in *wait* than DASH in async and so more time for the whole halo update process
(Fig. 6B). But, DASH lost time in *inner* and in *bound* (Fig. 6A).

In particular on *Haswell*, the halo update process was a large advantage for DASH; at
least by a factor of three (Fig. 6B) and could compensate the slower grid element update
process (*inner* and *bound*). A similar behavior can be seen on *Romeo*, but not as distinct
as on *Haswell* (Figs. 7B and 7A). The reason for the changed MPI behavior could be the
result of much more communication requests due to the strided access pattern. DASH had
one request per neighbor because of the buffered halo data which seems to be beneficial
for higher dimensions and large partitions.

In all weak scaling results DASH spent no significant time in *wait* as well as MPI in
*async*. Also, the amount of time spent for *bound* is very small compared to *inner*. To see
how these stages behave in a strong scaling scenario we used a squared structured grid
with $55000^2$ elements for 2D and $1500^3$ elements for 3D (fits into the main memory of one
compute node). The measurements were made on *Haswell* to involve as much compute
nodes as possible while still using all cores on these nodes.

Figures 8 and 9 show the results for the strong scaling scenario (2D and 3D). They
behaved quite similar in both scenarios. Until 16 cores on one node DASH outperformed
MPI, but on top of that, DASH and MPI had similar results (Figs. 8A and 9A). For the 3D
scenario with 16 and 32 compute nodes, DASH also outperformed MPI. Notably, both
exhibit the same performance artifact around 12 to 24 cores. The reason is an increased

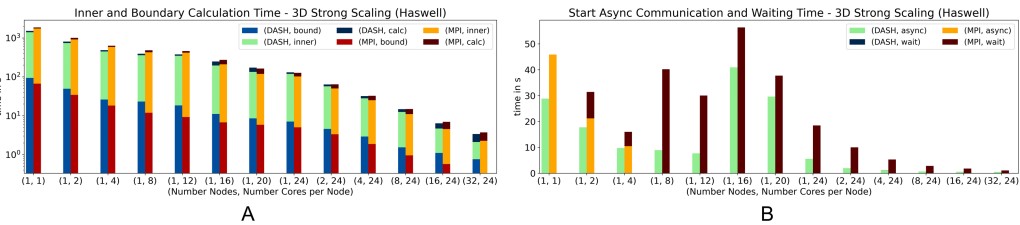

**Figure 9** Strong scaling for 3D heat equation—DASH *vs.* MPI on Haswell (values not stacked).

number of last level cache load misses and indicates that both implementations are memory bound at this number of cores per node.

Interesting, besides the total results, are the individual phases. The *inner* phase is processed faster by DASH in both scenarios, while *bound* is slower compared to MPI (Figs. 8A and 9A). The *HaloWrapper* has a slight disadvantage, by using separate memory for the halo elements. MPI instead is implemented with one contiguous memory block, including grid and halo elements. This can be better optimized by the compiler, in case the stencil points are accessed. In MPI, the stencil points can be accessed with a fixed offset on the *inner* as well as on *boundary*, while in DASH the stencil points in *bound* need to be requested for their location first to be correctly accessed afterwards.

Figures 8B shows a significant rise of the *async* runtime for 16 and 20 cores for the DASH implementation. This results from the halo element buffering. As mentioned before, the implementations suffered performance by many L3 cache load misses, which also applies for the halo element buffering described in the 'Internal management of halo memory and halo data exchanges' section. DASH still shows no significant runtime in *wait*, while MPI shows a notable runtime in *async* until four cores for the 3D scenario for the first time.

Overall, DASH can compete with the plain MPI example for all presented scenarios;by using less LOC. The HaloWrapper can also be easily adapted to a higher number of dimensions, other stencil shapes or grid border conditions.

### DASH vs. ScaFES

As mentioned in the 'Related Work' section, ScaFES is closest to our approach, so we also compared DASH with ScaFES. The heat equation example was implemented with ScaFES but with slight changes to the MPI scenario. Because ScaFES does not support connected global grid borders the heat equation is used with global boundary conditions instead. The DASH implementation used, therefore the CUSTOM GlobalBoundarySpecs. The detailed measurements of the stages *async*, *inner*, *wait*, *bound* were replaced by *update* (*inner* + *bound*) and *sync* (*async* + *wait*) because these two stages are already part of the internal measurements of ScaFES. The runtime measurements were done on *Haswell* for 2D and 3D. Instead of $55000^2$ elements for 2D and $1500^3$ elements 3D both implementations used $45000^2$ and $1300^3$ elements in the strong scaling scenario as total grid size, because ScaFES has a higher memory requirement than the MPI and DASH implementations. These grid sizes were also the basis of the weak scaling scenario. Furthermore, the number of iterations was reduced to 10 iterations, which were enough to show a different performance behavior.

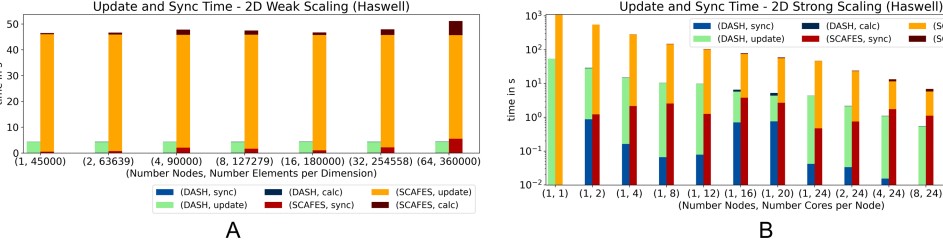

**Figure 10** Weak and strong scaling for 2D heat equation—DASH *vs.* ScaFES on Haswell (values not stacked).

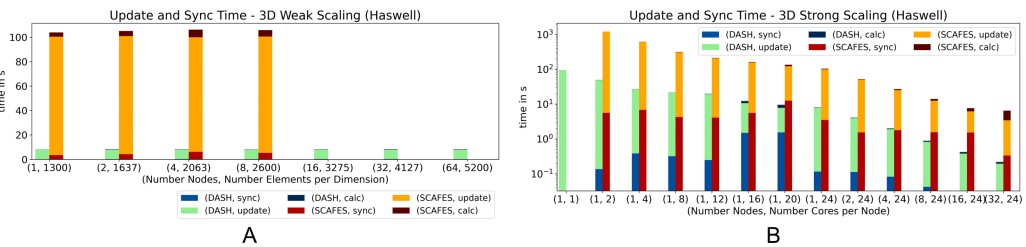

**Figure 11** Weak and strong scaling for 3D heat equation—DASH *vs.* ScaFES on Haswell (values not stacked).

Figure 10 shows the performance of DASH and ScaFES in a weak (Fig. 10A) and strong (Fig. 10B) scaling scenario. In both scenarios, DASH has a significant performance advantage over ScaFES. In particular, the update of the grid elements is up to a factor of 10 faster with DASH.

In case of 3D (Fig. 11) the big performance gap between DASH and ScaFES still exists in both scenarios. While in the weak scaling scenario (Fig. 11A the ScaFES runs aborted with segmentation faults for 16, 32 and 64 compute nodes, in the strong scaling scenario (Fig. 11B the single process ScaFES run also aborted with a segmentation fault. Interestingly all other process combination worked perfectly.

ScaFES mostly suffers from the coordinate based access pattern named ScaFES::Ntuple that need to be calculated for each grid element and all its neighbors. In DASH we optimized this form of access especially for the InnerIterator as described in the 'Stencil Point Access' section. It is also worth mentioning that the LOC of both implementations, DASH and ScaFES, are quite similar, as well as the additional LOC to switch from 2D to 3D. But in the case of ScaFES, the abstraction results in a significant performance loss. Both ScaFES implementations were reviewed by a former ScaFES developer to avoid performance decreasing mistakes.

## CONCLUSION AND FUTURE WORK

In this article we presented an abstraction named HaloWrapper for stencil based codes with halo areas for distributed n-dimensional structured grids on top of the distributed data

container NArray provided by the DASH C++ template library. The DASH HaloWrapper simplifies coding and code maintenance of parallel computations on distributed compute grids. It builds and manages the data distribution, halo environment, communication and data access based on user defined stencils and global boundaries conditions. For changes of the stencil shape, the number of dimensions or the grid size require only few code modifications. Compared to applications build up from scratch with *e.g.*, MPI, a user does not need deeper knowledge of parallel concepts or libraries such as MPI.

To measure the degree of abstraction (code complexity) and the performance we compared the HaloWrapper with a plain MPI counterpart and the C++ library ScaFES with a simple 2D and 3D heat equation example. ScaFES was interesting for us, because it follows the same design purpose and compared itself with PetSc in *Flehmig, Feldhoff & Markwardt (2014)*. The code complexity were measured in lines of code (LOC). The plain MPI solution needs significantly more LOC as the HaloWrapper and the ScaFES implementations, which are on the same level for the 2D implementation. The LOC gap between MPI and the HaloWrapper/ScaFES increases when higher dimensions are used. On the performance side, we compared the 2D and 3D implementations in a weak and a strong scaling scenario. The HaloWrapper and MPI implementations were on par with slight advantages for the HaloWrapper on average. The ScaFES based implementations instead were significantly slower than the HaloWrapper and plain MPI counterparts.

Thus, the advantages for simpler coding and its maintenance comes with no performance losses. Further additional optimizations like tasking, multiple threads, or the tuning of stencil operation are still possible on top of the presented approach and are primary goals for future work.

The source codes used in the 'Evaluation' section are available on GitHub (https://github.com/dash-project/dash-apps/tree/master/heat_equation) to view the code excerpts in their context and for reproducibility purposes.

## ACKNOWLEDGEMENTS

We would like to thank the members of the DASH team.

### Funding

This work was supported by the German Priority Programme 1648, Software for Exascale Computing (SPPEXA/DFG). The funders had no role in study design, data collection and analysis, decision to publish, or preparation of the manuscript.

### Grant Disclosures

The following grant information was disclosed by the authors:
German Priority Programme 1648, Software for Exascale Computing (SPPEXA/DFG).

### Competing Interests

The authors declare there are no competing interests.

## Author Contributions

- Denis Hünich conceived and designed the experiments, performed the experiments, analyzed the data, performed the computation work, prepared figures and/or tables, authored or reviewed drafts of the article, and approved the final draft.
- Andreas Knüpfer conceived and designed the experiments, analyzed the data, prepared figures and/or tables, authored or reviewed drafts of the article, and approved the final draft.

## Data Availability

The source code for the DASH and MPI implementation is available at GitHub: https://github.com/dash-project/dash-apps/tree/master/heat_equation.

The raw data is available in the Supplemental Files.

## Supplemental Information

Supplemental information for this article can be found online at http://dx.doi.org/10.7717/peerj-cs.1203#supplemental-information.

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
