# Peer review of "A Halo abstraction for distributed n-dimensional structured grids within the C++ PGAS library DASH"

_PeerJ Computer Science, doi:10.7717/peerj-cs.1203_

## Round 0.1 · original submission · Major Revisions

Please address all the comments of reviewers.

Reviewer 1 ·

Basic reporting

The article has some problems of the English used, several typographical errors and, I believe, is not sufficient self-contained to assess the relevance of the experimental evaluation. The authors assume that the reader has some prior knowledge of the DASH, DART and/or GASPI library.

Experimental design

There should exist additional experiments with developers to assess if the proposed abstraction is in fact better than the state of the art.

Validity of the findings

The data seems sufficient and valid. In the conclusions the authors should do a better effort in quantifying the gains of the proposed solution.

Additional comments

Summary:

This paper presents an abstraction to program distributed N-dimensional structured grids within an existing C++ library.
The authors show that such abstraction reduces the complexities of developing applications with the library.

Strong points:

The proposed abstraction seems to reduce the complexity of applications and there is some gains in terms of performance.

Weak points:

I am familiar with the concept of PGAS, but I am not familiar with the DASH library, nor with DART, nor with GASPI, as such I found a bit hard to understand the contributions of the authors. The only change is the API, right? Is there any relevant contribution at the level of the implementation? I believe the authors should provide a background section on how DASH implements some relevant datastructures. Section 3 presents only an overview of how the API can be used for a certain purpose, it is not enough to grasp how data is copied and processed, without that information is hard to understand the performance results in Section 5.2.

The authors should make clear what the contributions are somewhere in the introduction.

In Table 1 there is a difference of, e.g., 249-134 = 115 LOCs for the 2D application, but the parts in the following three columns only cover, 46 and 16, resp., MPI and DASH rows, what is missing? Is it relevant?

Comments:

I believe the authors have to conduct a study with developers (e.g., bachelor students) to better understand if the proposed abstraction is in fact less complex. E.g., via measuring the development time and the amount of errors in the produced code.

I did not understand this sentence (114-117): "In the end, the programmer still has to organize the update and access of halo elements (1, manage the adjacent partitions and combine the iterator access with the halo access." --> These seem quite a few tasks, I think it is relevant to have more details here or point to some (background) section.

On line 199, is "the wait call" step (4)? If so points (6) and (7) should be renamed to, e.g., 4a and 4b.

On the conclusion, line 336, the authors should quantify "[...] the slight advantage for the DASH [...]".

Typos:

The authors use negative contractions (e.g. can't, doesn't, isn't) extensively. I am not a native speaker, but I am used to not use them in technical documents. My advice to the authors is for them to look carefully into the following lines: 40, 69, 71, 84, 97, 112, Listing 2 (caption), 165, 173, 181, 187, 188, 224, 272 (there are others).

Line 116: "[...] a lot of code adaption [...]"

Table 1 and footnote url: eqation -> equation

In Section 5.2, MPI and DASH become mpi and dash, the authors should maintain some consistency.

Line 280: "[...] total caluclation time [...]"

Line 323: "[...] for all presented scenarios;by less LOC"

I think it is hard to read the figures in grayscale, the authors may want to review some of the colours used.

Line 338: "Furter optimizations [...]"

·

Basic reporting

The English throughout this article should be improved. I suggest reading back through the paper for grammatical mistakes, such as mixing singularity with plurality and capitalization errors. Examples of grammatical errors are included as highlights throughout the PDF, with additional small comments written on the pages.

There are a large number of existing stencil codebases, and this paper argues that theirs is novel because it uses one-sided communication. However, this paper does not compare against any existing codebases or discuss the benefits of one-sided communication.

The paper is difficult to follow. I suggest using one example throughout the paper. When discussing each part of the new library, the authors could relate the concepts to this example. Without an example to follow, the large number of definitions (e.g. StencilIterator, InnerIterator, BoundaryIterator, BoundaryRegion, StencilSpecs, StencilOperator, …) are hard to understand.

Experimental design

There are a large number of existing stencil codebases, some of which are mentioned in the related work section. This paper presents a new library for stencil computations that uses one-sided communication. However, results only compare against a basic MPI version rather than an existing library. How many changes are needed to convert from 2D to 3D in other existing libraries?

It is unclear what benefit is provided by using one-sided communication. If one-sided communication outperforms MPI for stencil computations, I suggest highlighting this in the results.

The description of the MPI implementation of stencil computations is unclear. The paper states that many MPI requests are required as memory is not contiguous, but standard MPI stencil computations either pack to a contiguous buffer or use an MPI data type. However, the provided GitHub code only sends a single MPI message per direction. This section does mention that the authors copy to a contiguous buffer, but it sounds as if this is a part of the DASH library as an MPI implementation for comparison was not discussed before the results.

Validity of the findings

The authors present a library that requires fewer code changes than writing a stencil computation within MPI from scratch. Furthermore, the authors show that this library does so without harming the performance. However, there are no comparisons to existing libraries to verify whether the presented library improves over an implementation that is accepted as performant. I suggest comparing against libraries such as ScaFES and PETSc. Furthermore, I suggest highlighting the benefits of one-sided communication.

---

## Round 0.2 · accepted · Accept

While in production you may choose to also address the remaining minor comments from Reviewer 2 in order to improve the quality of your paper.

Reviewer 1 ·

Basic reporting

The paper has improved since the last submission regarding the text quality and I believe that is now closer to publication.

Experimental design

The experiments are now more detailed and are compared against one more library, which improves the contributions of this paper. However, statements like "[...] ScaFES runs aborted with a segmentation faults for[...]. Interestingly all other process combination worked perfectly." I wonder whether ScaFES experiments worked due to "luck", say there is some buffer overflow or one is using uninitialized memory. If ScaFES does not work I would suggest the authors to look for other competitor or to assess (debug) better why/where the segmentation fault occurs. Given that a ScaFES author reviewed the code, I would ask them why these faults occur. If the reason is due to the complexity of setting up ScaFES it would be a compelling argument to use HaloWrapper instead.

Validity of the findings

I have not replicated the results, but I believe it is possible to replicate given that one has access to the repository with the code and equivalent machines. The data seems sound and supports the final conclusions.

Additional comments

I would appreciate a change on the text where is stated that ScaFES crashes by means of adding more information on the matter, but do not oppose the paper to be accepted as is.

·

Basic reporting

Line 73 : adpoting should be adopting

It would be helpful if the authors listed their contributions. I am a bit confused by the rebuttal statement that "our main goal was to show, that our abstraction doesn’t lose to much performance compared to a plain MPI solution".

Many of the bar plots have colors that are not visible (left hand plots in Fig 7). The extra colors seem to add complexity for no real benefit unless the authors are trying to make the point that MPI doesnt have strong progess and time is spent in the wait instead of elsewhere. If this is an important point it would be nice to have that highlighted, but otherwise the extra complexity in the plot seems unneeded.

Experimental design

no comment

Validity of the findings

no comment